# Assessing Options for Remediation of Contaminated Mine Site Drainage Entering the River Teign, Southwest England

Abigail Jordan , Rachel Hill, Adrienne Turner, Tyrone Roberts and Sean Comber *

School of Geography, Earth and Environmental Sciences, University of Plymouth, Drakes Circus, Plymouth PL4 8AA, UK; abigail.jordan@postgrad.plymouth.ac.uk (A.J.); rachel.hill-1@students.plymouth.ac.uk (R.H.); adrienne.turner@students.plymouth.ac.uk (A.T.); tyrone.roberts@students.plymouth.ac.uk (T.R.)
* Correspondence: sean.comber@plymouth.ac.uk

**Abstract:** The river Teign in Devon has come under scrutiny for failing to meet environmental quality standards for ecotoxic metals due to past mining operations. A disused mine known as Bridford Barytes mine, has been found to contribute a significant source of Zn, Cd and Pb to the river. Recently, studies have been focused on the remediation of such mine sites using low-cost treatment methods to help reduce metal loads to the river downstream. This paper explores the metal removal efficiency of red mud, a waste product from the aluminium industry, which has proven to be an attractive low-cost treatment method for adsorbing toxic metals. Adsorption kinetics and capacity experiments reveal metal removal efficiencies of up to 70% within the first 2 h when red mud is applied in pelletized form. Further, it highlights the potential of biochar, another effective adsorbent observed to remove >90% Zn using agricultural feedstock. Compliance of the Teign has been investigated by analysing dissolved metal concentrations and bioavailable fractions of Zn to assess if levels are of environmental concern. By applying a real-world application model, this study reveals that compressed pellets and agricultural biochar offer an effective, low-cost option to reducing metal concentrations and thus improving the quality of the river Teign.

**Keywords:** trace metals; mine remediation; zinc; red media; biochar

## 1. Introduction

Historic mining in the southwest of England has left a legacy of environmental and socio-economic impacts. Whilst mining operations have largely ceased throughout Devon and Cornwall, impacts have persisted, resulting in localised contamination and elevated metal concentrations in soils, sediment, and waters. In England, pollution from mine waste affects over 1700 km of rivers [1] with the potential to reduce the quality of drinking water and threaten sensitive aquatic ecosystems. This legacy presents a challenge in achieving the requirements set out by the Water Framework Directive (WFD) (Directive 2000/60/EC) which has established environmental quality standards (EQS) for specific pollutants such as arsenic (As), zinc (Zn), copper (Cu), iron (Fe), chromium (Cr) and manganese (Mn), priority substances such as lead (Pb) and priority hazardous substances such as cadmium (Cd). Meeting the standards and protecting the quality of our water bodies are therefore of fundamental importance.

The river Teign, sourced in Dartmoor, Devon, is at risk of not meeting the requirements set out by the WFD and forms the focus of this study. Exploitation of mineral resources at a local disused mine in Bridford, known as Bridford Barytes mine, has contributed to elevated concentrations of potentially toxic metals. Mining for baryte (barium sulphate) took place between 1855 and 1958. However,

prio to this, Pb-Zn mining occurred within the catchment [2]. Both episodes have been responsible for releasing potentially ecotoxic metals into the river Teign and monitoring data have consistently shown exceedances in metal concentrations, particularly Zn, which presents the basis for this investigation.

Metals sourced from mining operations are typically discharged from mine adits, where acid mine drainage (AMD) is generated, releasing trace metals into the environment with potentially adverse effects on the ecology. AMD is produced when sulphide-bearing minerals released from mining activities are exposed to atmospheric conditions. The most common sulphide mineral in this process is pyrite ($FeS_2$). The oxidation of pyrite leads to the generation of sulphate and an increase in proton acidity [3]—this reaction is responsible for considerable increases in acidity within the natural environment. Due to this increase in acidity, the pH associated with AMD is typically below 4.0 [4], in which metals are highly soluble and easily mobilised—commonly these metals include Mn, Cr, Cd, Zn, Pb and As.

Zn is one of the most encountered WFD-specific pollutants from mining activities. It is a metal both essential and toxic to organisms, and monitoring the concentration of Zn at the catchment scale is therefore critical to help sustain and preserve the environment. Notably, Zn is often present in high concentrations due to the background geology; this presents unique complications in assessing the risk of impacts. However, studies have shown that Zn in sediments of the river Teign and estuary is not entirely naturally occurring and is derived from mining pollution [2]. These elevated levels of Zn have been attributed to the episodes of Ba and Pb-Zn mining throughout the Teign catchment including a major source at Bridford Barytes mine. In catchments affected by AMD, Zn is commonly present in its most ecotoxic form $Zn^{2+}$, as is the case with the river Teign. The release of this hydrated Zn ion into the environment is toxic to aquatic biota at elevated concentrations, with reports of reproductive and developmental responses in fish and other aquatic organisms [5]. With regards to human health, long-term excessive exposure has been identified as a contributing factor to chronic diseases, a decrease in immune system function and even infertility [6,7]. Preventing such adverse effects to aquatic life and human health is the driving force behind environmental legislation.

Over the years, growing concern for the environment and human health has led to an increase in legislation governing pollution associated with the mining industry. The WFD has become one of the most influential pieces of EU law concerning water pollution and the quality of our water bodies. The directive is built upon the principles of sustainable development and requires the development of management strategies referred to as River Basin Management Plans (RBMPs). It also requires member states to classify the ecological quality of waters once every 6 years as either high, good, or moderate; with pass/fail environmental quality standards (EQS) for chemicals of concern. To achieve good status, all the chemical and ecological parameters have to be 'good', as it is a one out–all out assessment. Currently, the lower Teign catchment only achieves 'moderate' status for failing to meet the standards required for good ecological classification and for periodic failures of the Zn EQS [8]. Consequently, understanding the contribution of Zn to this river catchment and undertaking appropriate mitigation are key to meeting the demands of the WFD and are the rationale behind this study.

The EQS established by the WFD are based upon recommendations from the United Kingdom Technical Advisory Group (UKTAG) and are monitored closely by the Environment Agency. They are derived from present scientific understanding of the conditions needed for a healthy water environment and utilise ecological data from thousands of sites across the UK. The revised standard for Zn in freshwaters is currently 10.9 μg/L bioavailable plus the ambient background concentration 2.9 μg/L [9]. Importantly, the chemical form of zinc is greatly influenced by the hydrological and physiochemical conditions of the water [10]. Metal concentrations, pH conditions and amount of organic matter all control the bioavailability and toxicity of zinc [11], not considering the bioavailable fraction of zinc may result in an under or over estimation of the risks posed by the metal. These influences are therefore an important consideration when assessing whether a water body is in fact 'failing' due to the presence of the metal and is consequently an environmental concern.

Practical and cost-effective treatment for mine water is topical and extensive research has been undertaken to assess the application of different treatment methods in the UK. The degree of environmental pollution generated by AMD is highly variable, meaning treatment must be flexible and specific to each site. Passive methods to remove heavy metal ions are currently favoured due to their low cost and local availability, with techniques including constructed wetlands, limestone for neutralisation, precipitation, and adsorption [12]. Biochar is an attractive, low-cost, adsorbent material whose adsorptive properties can be influenced by the type of feedstock used. The remediation potential of biochar has been noticed by previous studies [13,14], with focus on the effects of pyrolysis temperature, contact time, initial metal concentration and type of feedstock used. It has been found that agricultural biochars have high adsorption capacities (11,000 mg/kg) compared to wood biochars (395.8 mg/kg) [15,16]. This study emphasizes the significance of using different biochar feedstock and their influence on the removal of Zn, Cd and Pb.

Red Media Technology has been trailing the capability of 'red mud' (RM) for adsorbing heavy metals from discharged mine waters at a relatively low cost. Millions of tonnes of hazardous RM waste is produced each year as a by-product of the aluminium industry; the utilisation of this material therefore supports the concept of waste recycling. It is a highly alkaline material with a pH of 10–13, and the red colour comes from the presence of oxidised iron, which comprises up to 60% of the mass of the product [17]. The RM is in pellet form, and pre-treatment of the pellets via heat and acid treatment has been found to increase adsorption and the removal efficiency of heavy metals [18]. Laboratory studies have investigated the capabilities of four different types of pellet which have undergone treatment: compressed (CP), fired (FP), fired acid etched (FAE) and a new powdered pellet (PP). Importantly, studies have shown that the pre-treatment of pellets is essential in the adsorption process and hence determines the overall effectiveness of removing heavy metals from mine water [19]. However, they seldom consider the practicalities of applying these treatment methods to a real-world application. This report aims to evaluate the feasibility of red mud pellets and biochar as treatment methods, weighing up the benefits and costs to determine which method will be most applicable for reducing metal loads to the Teign. Principally, this report focuses on Zn. However, the removal efficiencies for the priority substances Cd and Pb have also been considered for comparison.

Compressed pellets have been tested in the laboratory and during a field-scale trial by Hill (2016) [19] and Comber (2015) [20], respectively. CP have been compacted under high pressure, forming small and crumbly pellets of varying sizes (Figure 1a) [19]. They have lost porosity during compaction and have a high surface area; however, the field trial shows that the pellets lack structural integrity and suffered degradation during the experiment [20]. Lab-based experiments using fired pellets have been conducted by Hill (2016) [19] and Turner (2017) [21]. Production of the FP involves heating in a kiln at 1050 °C for 2 h and cooling for a further 2 days [19]. The pellets are more uniform in size, with a coarse texture and an overall lower surface area compared with the compressed pellets (Figure 1b) [21]. The adsorption efficiency of the fired acid etched pellets has been tested by Turner (2017) [21], where they are described as small, bright orange pellets with a powdery texture and a smoother surface produced from etching (Figure 1c). The powdered pellets described by Turner (2017) [21], are similar in appearance to the compressed pellets, with a cylindrical shape, powdery texture, and a dark orange colour, fired at 800 °C.

Ultimately, information collated on these studies of different treatment methods are employed by the competent authorities (Environment Agency and Coal Authority) to develop and build mine water treatment schemes to clean up our waters where the quality has been compromised by pollution from abandoned mine sites [1]. Currently, one of the greatest challenges in treating pollution generated from AMD is finding a method that meets the expectations of efficiency, cost, and sustainability. The objective of this study is to assess the necessity of Zn, Cd and Pb removal in the river Teign and evaluate the efficiency of treatment methods that utilise red media. The results will enable an assessment of the practicalities associated with reducing Zn loads to the catchment, and an overall more comprehensive understanding of adopting low-cost adsorption treatments to mine sites in the UK.

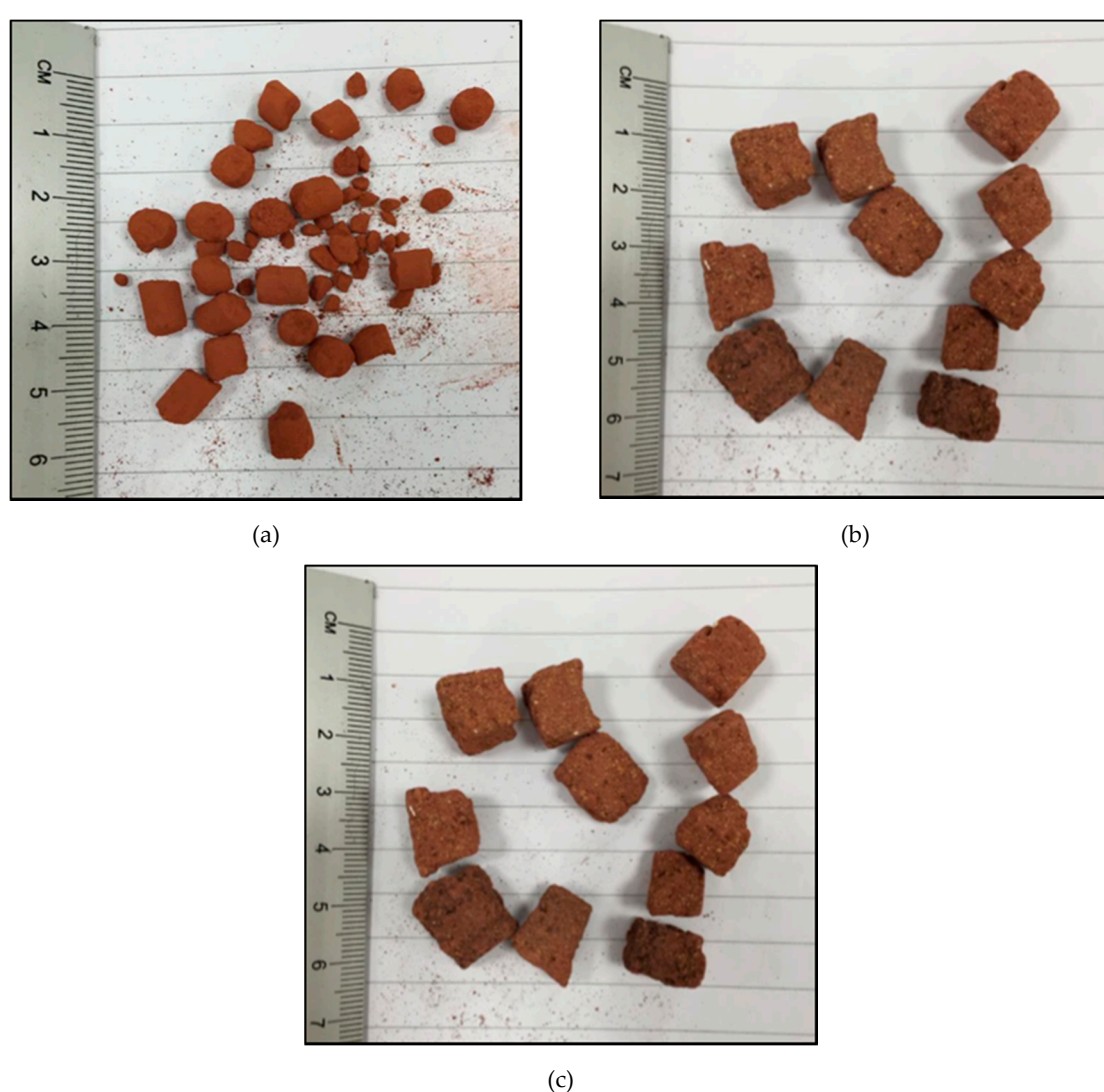

**Figure 1.** Red media pellets which have all been pre-treated. (**a**) Compressed pellets [19]. (**b**) Fired pellets [19]. (**c**) Fired acid etched pellets [21].

## 2. Methodology

### 2.1. Study Area

This study is based on a former baryte mine in Bridford, situated southwest of Exeter (SX83148643). The mine is located within the Teign valley on the north eastern edge of Dartmoor (Figure 2) where metalliferous mineral deposits have been extracted since the bronze age due to the presence of a large granite batholith. Mineral deposits in the area consist mainly of shales, mudstones, cherts and tuffs, also known as the Culm measures; these deposits contain the Ba-Pb-Zn loads [22]. Initially, Pb mining took place at the site dated at approximately 1804. However, low profits moved production to Ba in 1855, with final abandonment of the mine in 1958 [2,20].

Mine water discharged from the main adit is channelled to the Bridford beck via an Environment Agency monitoring point. The Bridford beck is a tributary of the Rookery brook sourced in Dartmoor which flows downstream approximately 1 km into the river Teign, and both water courses currently exceed the Zn EQS [19,20]. However, these tributaries comprise a small area of the catchment (approximately 6 km$^2$ out of 540 km$^2$ for the Teign catchment [23]) and, at first instance, seem unlikely to contribute greatly to the elevated Zn concentrations of the Teign.

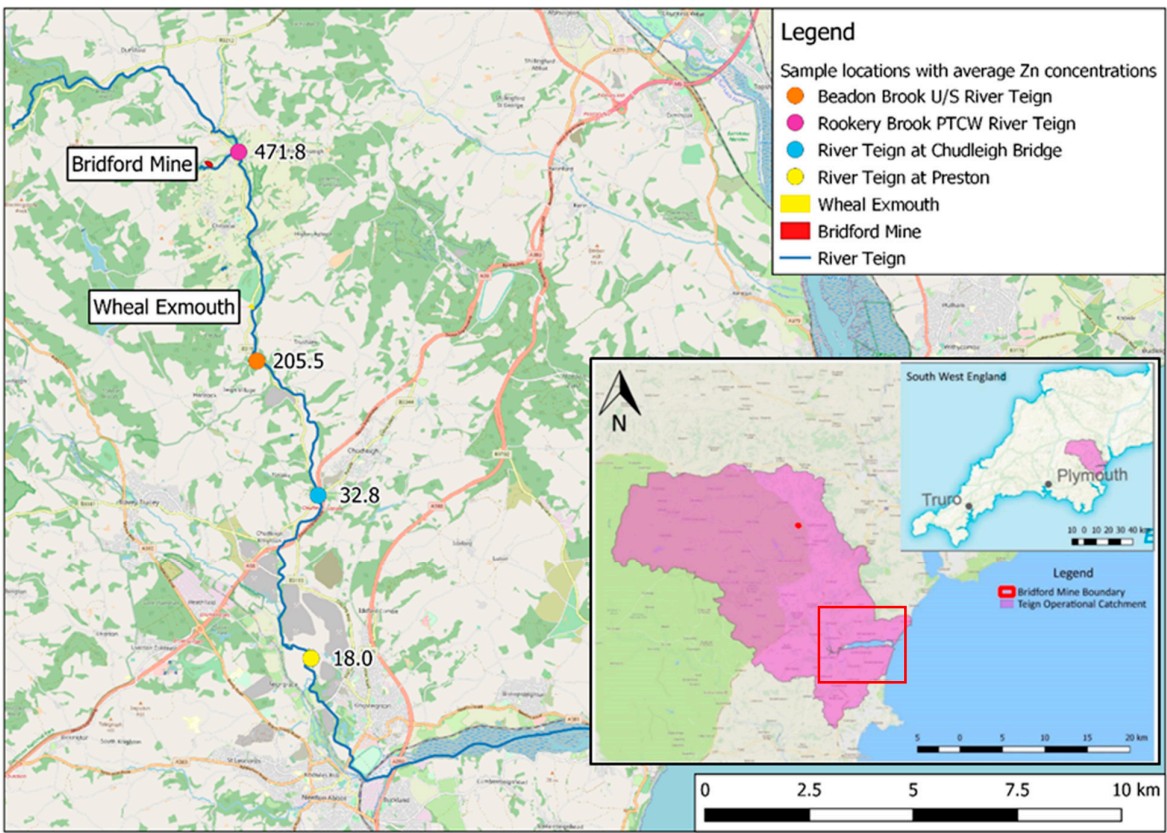

**Figure 2.** Map showing the location of Bridford baryte mine which is situated on the north eastern edge of Dartmoor within the Teign catchment along with the Environment Agency sampling points and mean zinc concentrations (µg/L) (2000–2020 data).

## 2.2. Current Studies Using Red Media Technology Products

Laboratory studies have been undertaken to assess the removal efficiency of pre-treated pellets [19,21]. Samples were collected from the adit outflow in June and November 2016 at Bridford along with in situ measurements of pH, temperature, dissolved oxygen content and redox potential. Previous monitoring has shown concentrations of trace metals in the adit discharge to be remarkably stable over time. CP, FP, FAE and PP were supplied by Red Media Technologies to determine their metal removal efficiency and suitability to a mine environment [24]. An adsorption kinetics experiment tested the rate of analyte adsorption by adding 850 mL of mine water to 200 g of RM pellets in a 1 L polythene bottle followed by continuous agitation on an orbital shaker, with 9 mL of sample being remover by syringe at set time intervals which were filtered through cellulose nitrate 22 mm membranes before preservation using ultra pure nitric acid (100 µL of 20% acid). A full outline of this methodology can be found in the Electronic Supporting Information (ESI, Figure S1). Starting and final analyte concentrations of the mine water were analysed using Inductively Coupled Plasma instruments such as Inductively Coupled Plasma Mass Spectrometry (ICP-MS, Thermo Scientific X Series 2, with indium and iridium internal standards, Thermo Fisher Scientific, Waltham, MA, USA) and Inductively Coupled Plasma Optical Emission Spectrometry (ICP-OES; Thermo Scientific ICAP 7400 Series with yttrium internal standard, Thermo Fisher Scientific, Waltham, MA, USA), and the removal efficiencies were then calculated after 2 h for Cd, Pb and Zn. Briefly, ICP-OES was used for samples with metals in the mg/L range and ICP-MS for metals in the µg/L range. Certified reference materials (Enviromat, EPL-3), internal control samples and blanks were determined within each batch of samples to ensure data quality. Recoveries for Zn, Pb and Cd were 100% ± 10% and precision for the 3 replicate analyses for each sample were

typically less than 5% relative standard deviation [19,21]. The pH was also tested at the start and end of the experiment to reveal any neutralising capabilities of the pellets [21].

As well as a kinetics experiment, an adsorption capacity column experiment was undertaken to determine the adsorption behaviour and optimum capacity of the CP and FP in mg of metal sorbed/kg of media used. Depending on the amount of RM material available, columns were either clear polycarbonate with approximately 1 L capacity or a 60 mL polythene syringe. The columns were bunged at either end with fittings to accept 1.5 mm diameter polythene tubing from a Gilson Miniplus 3 peristaltic pump. The columns (3 replicates) were packed with test material and mine adit water passed through at a rate of typically 1 mL/min. Adit water exiting the column was collected (9 mL), filtered and preserved as per the kinetic experiment. The adsorption capacity was calculated using the starting concentrations of the elements, the amount of solution which had flowed through them and the weight of pellets within the column (ESI, Figure S2). The highest capacity achieved for each metal has been recorded [19].

A field-scale trial of the removal efficiency of toxic metals using the pellets was undertaken by Comber (2015) [20] in conjunction with Red Media Technology at Bridford Barytes mine, Bridford. The trial period was a duration of 3 months to assess the performance of the pellets on a realistic timescale. The experiment consisted of a 1 m$^3$ tank containing compressed pellets (CP); mine water was delivered to the tank and samples were taken throughout the operation, including pH readings. Minewater was delivered to the test rig using a peristaltic pump with flexible pipework from the adit discharge point. The flow rate into the test rig was initially set at approximately 15% of the mine discharge flow and was adjusted to ensure consistent flow through the media tank (110 l/hr). An initial residence time of one hour was altered as the trial continued, so as to give data for additional hourly intervals up to 8 h residence time. Metal concentrations were determined by ICP-MS as described above. Samples were collected from the inlet and output from the tank, filtered and preserved as for the laboratory studies.

Analyte concentration data collected from the laboratory tests and field-scale trials have been used to calculate the metal removal efficiency of each pellet form, as well as the adsorption capacity and pH neutralising capability; the results will allow an evaluation of which pellet is most suitable for reducing the Zn load from Bridford to the river Teign.

### 2.3. Alternative Treatment Method Using Biochar

Biochar is a black, carbon-rich solid produced by thermal decomposition of biomass, similarly to charcoal. Typically, it has a wide range of characteristics which depend upon the feedstock used; this affects the chemical and physical properties of the biochar and consequently how it acts as an adsorbent. The test data described here [25] quantified the sorption capabilities of pelletized biochar supplied by the United Kingdom Biochar Research Centre (UKBRC, Edinburgh, UK). Varying forms of feedstock were tested at different pyrolysis temperatures (550 °C and 700 °C) including char produced from forestry waste, municipal waste, and agricultural waste. Following a similar methodology to the experiments for the red media study, adsorption rates and adsorption capacities were determined for the same mine adit water. Metal concentrations were analysed by ICP-MS and ICP-OES as described above.

### 2.4. River Teign Metal Concentrations

The Environment Agency (EA) act as the competent authority to implement the requirements set out by the WFD and closely monitor the quality of water courses within England. Data provided by the EA's water quality archive have been extracted to determine the mean concentrations of the river Teign for dissolved Zn from 2000 to 2020. Analysing total dissolved metal concentrations forms the first stage of a tiered approach to assessing the classification of a water body in the UK [10], and if the Teign exceeds the standard EQS value of 13.8, then it will progress to the next tier. Bioavailability data are accessible after 2015 from EA monitoring data, and identification of the bioavailable concentration

of the metal allows direct comparison with the bioavailable EQS (10.9 µg/L for zinc) and forms the second tier for assessing compliance of the Teign with the WFD.

Several sample locations along the course of the river Teign have been selected to represent changing dissolved metal concentrations downstream from Bridford mine (ESI, Figure S3). Closest to the mine adit is the Rookery brook tributary, which flows into the Teign. Further downstream east of Canonteign is the Beadon brook past Wheal Exmouth mine site. Discharges sourced from Bridford mine and Wheal Exmouth are intercepted by the Teign at Chudleigh bridge. Dissolved concentrations of Zn, Cd and Pb were obtained for these selected sites from the EA, as well as some bioavailable data for Zn, calculated using the physiochemical parameters dissolved organic carbon (DOC), pH and Ca/Hardness (ESI, Figure S4). Notably, concentrations of Cd and Pb were frequently below the limit of detection (LOD), particularly from 2000 to 2010, and these results therefore have a high uncertainty.

### 2.5. Real-World Application Model

Using mean bioavailable metal concentration data and flow data from the Chudleigh river gauging station available from the National River Flow Archive, the average load of Zn, Cd and Pb into and within the river Teign at Chudleigh has been calculated using a simple spreadsheet model [19] (ESI, Figure S5), which simply combined flows and concentrations from the mine adit with river data (flow and concentrations) in order to generate loads of the trace metals entering the river and therefore the mine adits contribution. River metal concentrations were available online from the Environment Agency's Water Information System. Combining the EQS for the metal within the river with the flow provided a 'target' metal load to be achieved. The actual load was calculated by multiplying the latest monitoring concentration data by the flow. Subtracting the 'target' metal load for EQS compliance from the current load generated a load of metal required to be removed from the adit flow. It was then a simple case of using the pellet metal adsorption capacities to estimate the amount of pellets per year (tonnes) required to reach the EQS for the water quality monitoring point at Chudleigh.

## 3. Results

### 3.1. Removal Efficiency Results

Zn concentration data at Bridford mine adit are presented in Figure 3. The results show Zn concentrations over a 24 h period influenced by the different pellet forms.

The PP show a steady decline in Zn concentration within the first 2 h from 11,600 to 8900 µg/Lat 24 h, the final concentration is 616 µg/L. The FAE pellets show a similar trend in the first 2 h, with concentrations falling from 11,300 to 8410 µg/L t 24 h, Zn concentration is 311 µg/L. The CP exhibit the steepest decline in Zn concentration and hence the fastest removal rate, with concentrations decreasing from 9643 to 575 µg/L in just 2 h. Concentrations fall to 43.8 µg/L at 24 h. Finally, the FP show the slowest decrease in Zn concentrations within the first 2 h (9643–1388 µg/L). However, afterwards, concentrations rapidly decline to 68.9 µg/L at 24 h. These results reveal that the adsorption efficiency is strongly influenced by the pre-treatment of the pellets.

Levels of pH of the mine water during the experiments show that all the pellets have neutralising capabilities and produce alkaline conditions (Table 1). The CP and FP have a greater pH increase compared with the other pellets, particularly the FP, which have the largest pH increase of 4.68. A limitation of the pH test is that data for the FAE and PP pellets were recorded at a shorter duration of 6 and 24 h, respectively.

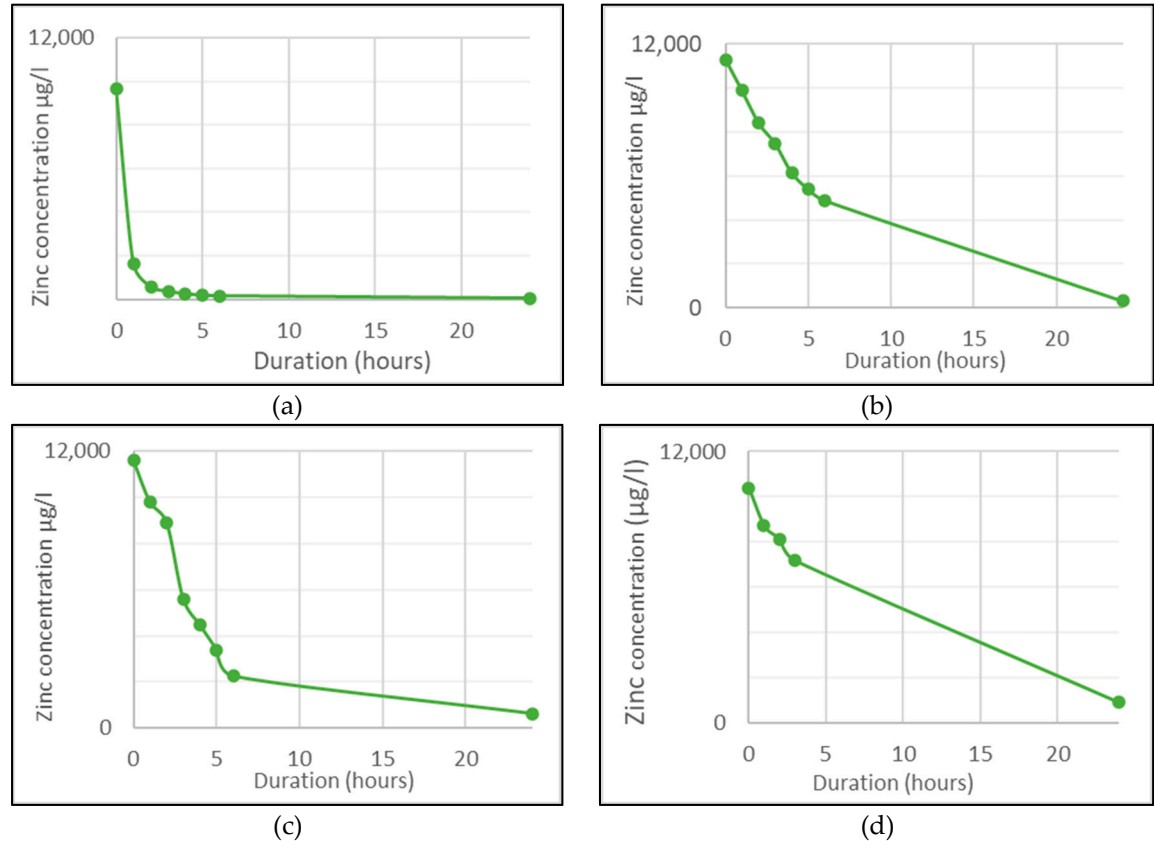

**Figure 3.** Zinc concentrations over a 24 h period influenced by the different pellets. (**a**) Compressed pellets (CP), (**b**) fired acid etched pellets (FAE), (**c**) powdered pellets (PP), (**d**) fired pellets (FP).

**Table 1.** PH changes in the mine adit water using the different types of pellet. Raw data extracted from Turner (2017) [21] and Hill (2016) [19].

| Duration | pH of Mine Adit Water | | | |
|---|---|---|---|---|
| | Compressed Pellets (CP) | Fired Pellets (FP) | Fired Acid Etched Pellets (FAE) | Powdered Pellets (PP) |
| **Start of Experiment (0 h)** | 4.65 | 4.65 | 3.78 | 4.59 |
| **End of Experiment** | 7.80 (53 h) | 9.33 (53 h) | 5.5 (6 h) | 8.84 (24 h) |

The removal efficiencies for Zn have been calculated using starting and final Zn concentrations and are summarised in Table 2 (ESI, Table S1). The results show the CP to have the fastest rate of removal for Zn within the first 2 h, achieving a high removal efficiency at 53 h. The FP achieve the highest removal efficiency at 53 h despite the slowest decrease in Zn concentrations at the beginning of the experiment. The PP and FAE pellets show slightly lower removal efficiencies than the other pellets at 24 h, but still reach a removal efficiency of 95%+.

**Table 2.** Efficiency of the different pellets for adsorbing Zn at 2, 24 and 53 h [19,21].

| Hours | Removal Efficiency for Zinc (%) | | | |
|---|---|---|---|---|
| | Compressed | Fired | FAE | Powdered |
| **2** | 73.7 | 22.0 | 25.6 | 23.3 |
| **24** | 99.5 | 99.3 | 97.2 | 94.7 |
| **53** | 99.8 | 99.9 | - | - |

Results from the adsorption capacity column experiment are shown in Table 3. Limited experiment duration meant that the highest capacity achieved was calculated using the starting and final concentration of the analytes, the weight of the pellets (CP 589 g), (FP = 410 g), and the amount of liquid flowing through (1 mL/min) [19]. Due to the adsorption capacity not being sufficiently reached in this experiment, more realistic capacities for the CP were used from the field-scale trial by Comber (2015) for the real-world application model.

**Table 3.** Highest adsorption capacities achieved from the column experiment for the CP and FP [19].

| Highest Adsorption Capacity Reached | Adsorption Capacity of Pellets (mg/kg) | | |
| :---: | :---: | :---: | :---: |
| | Zn | Cd | Pb |
| Compressed Pellet | >105.6 | >1.1 | >5.36 |
| Fired Pellet | >150 | >1.56 | >3.89 |
| Field-Scale Trial (Compressed Pellet) | 8743 | 35.40 | 2089 |

The high removal efficiency of the CP is supported by the field-scale trial conducted by Comber (2015) [20]. Figure 4 shows the removal efficiency of the pellets over a 3 months period. Influent concentrations of metal were relatively stable, varying by only up to 11% (relative standard deviation) for the different metals. The results reveal that >80% of the Zn is removed within the first 10 days of the experiment. After this period, the removal efficiency gradually falls until it remains at below 40% after 40 days. Cd and Pb follow a similar trend, but with a marked increase in removal after 70 days. These results suggest that the RM pellets require at least 2 h to be efficient and achieve >70% removal, uptake is reduced greatly up to 24 h and beyond (ESI, Table S2).

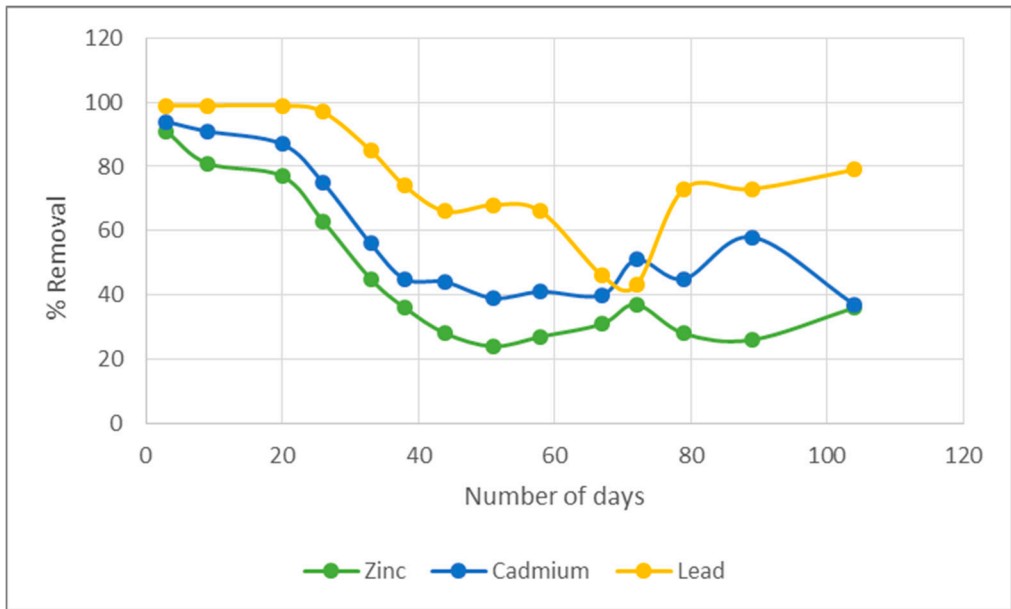

**Figure 4.** Graph showing results from the field-scale trial. Removal efficiency of the compressed pellets for adsorbing filtered zinc, cadmium and lead over a 3 month period at Bridford Barytes mine. Data taken from Comber (2015) [20].

When compared with the priority substances Cd and Pb (Figure 5), Zn appears to have the most similar adsorption rate to Cd, which is highest when influenced by the CP and lowest with the FP and FAE pellets. The results show Pb to have the greatest removal compared to Cd and Zn with all the pellets, especially the CP, which exhibit nearly 100% removal efficiency. Notably, the data only show

results for a 2 h duration; the FP are recognized to have the slowest removal efficiency during this time despite the greatest overall removal efficiency, as demonstrated by Figure 3 [21].

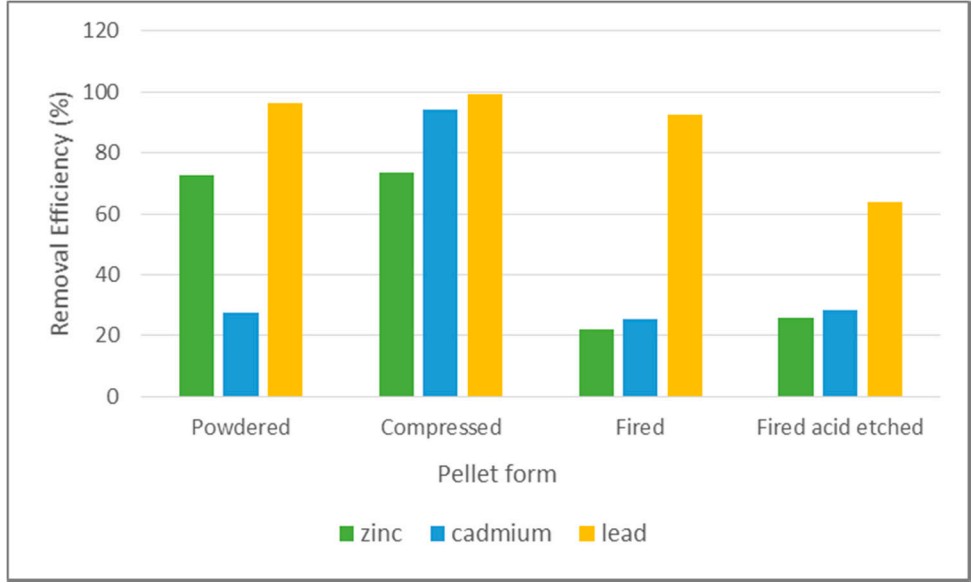

**Figure 5.** Column chart showing the removal efficiencies of Zn, Cd and Pb with the different pellet types over a 2 h period.

The results for the different biochar feedstock are shown in Figure 6 at pyrolysis temperatures of 550 and 700 °C (ESI, Table S3). Overall, the feedstock with the highest removal efficiency after 2 h is the agricultural waste (Miscanthus straw pellet, wheat straw pellet and oil seed rape) with analyte removal of over 80%. Forestry waste and municipal waste have the lowest removal efficiency compared to the other types of feedstock. Pb is the most effectively removed analyte, with a removal rate of >90% for the agricultural waste, whereas Zn has the lowest removal efficiency for all the biochar feedstock. Notably, there is no real difference between the removal efficiency at pyrolysis temperatures of 550 and 700 °C, except lead has a slightly higher removal efficiency at a temperature of 700 °C.

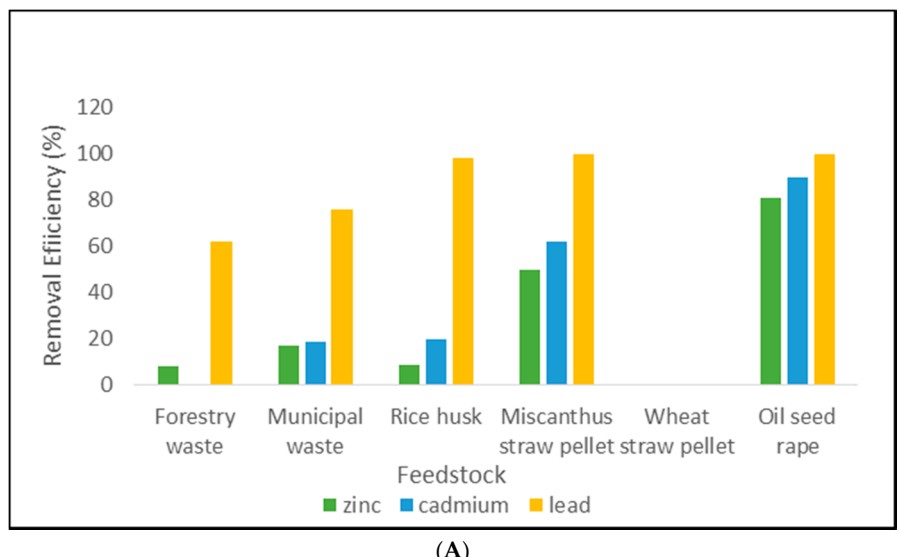

(**A**)

**Figure 6.** *Cont.*

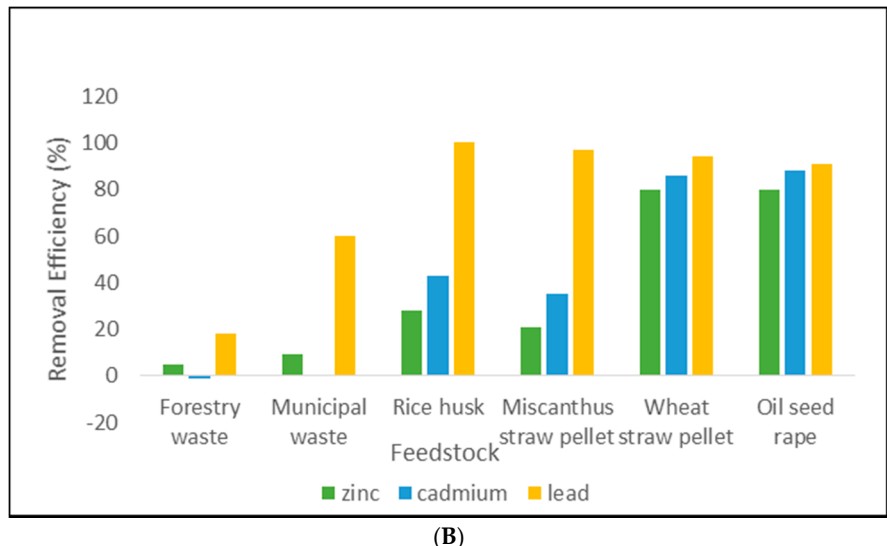

(**B**)

**Figure 6.** Column charts showing the removal efficiency of biochar at pyrolysis temperatures of (**A**) 550 °C and (**B**) 700 °C for Zn, Cd and Pb [25].

### 3.2. River Teign Metal Concentration Results

Selected sample locations downstream of Bridford mine are presented in Figure 1, with average dissolved Zn concentrations at each locality calculated from 2000 to 2020. Estimates of Zn concentrations at the adit have been made: 11,170 µg/L [26], 8911 µg/L [19], 11,200 µg/L [21] and 11,400 µg/L [20]. These values show that the adit acts as a point source of consistently high Zn values of approximately 11,000 µg/L. Downstream of the adit, mine waters enter the Rookery brook where average Zn concentrations are 471.8 µg/L, this is considerably higher than upstream values of 49 µg/L documented by Hill (2016) [19], owing to the mine discharge from Bridford. Further downstream, Zn concentrations are reduced to 205.5 µg/L at Beadon brook and then to 32.7 µg/L at Chudleigh bridge. Whilst Zn levels are observed to decline downstream, they remain above the EQS of 13.8 µg/L throughout the course of the Teign before entering the lower estuary where levels are reduced to 4.8 µg/L.

Dissolved concentrations of Zn, Cd and Pb recorded at the sample locations have been collated to show the changing metal concentrations between 2000 and 2020 (ESI, Table S4); the results are presented in Figure 7. The Rookery brook prior to confluence with ((PTCW) river Teign) data show Zn levels to initially be declining followed by a slight upward trend after 2013. Concentrations still greatly exceed the EQS, with levels of 407 µg/L in 2016, almost 30-fold the EQS. Cd and Pb show a similar trend of levels greatly exceeding the EQS despite an overall decline in recent years. Downstream at Beadon brook, Zn concentrations fluctuate yet show a general decline to 138.9 µg/L in 2018; this is still 10-fold above the EQS. Cd levels are comparable to Zn with declining concentrations of 1.64 µg/L in 2018, 20-fold above the Cd EQS of 0.08 µg/L. Meanwhile, Pb levels remain consistently below the EQS of 7.2 µg/L, with levels recorded at 1.71 µg/L in 2019. Further along the river Teign at Chudleigh bridge, metal concentrations are significantly lower than the previous localities. However, dissolved Zn remains above the EQS and shows rising levels since 2014 to 42 µg/L in 2019; this is still 3-fold the EQS. Cd follows a similar trend but appears to be steadily declining in recent years to 0.19 µg/L, 2-fold the EQS. Pb levels, however, continue to stay below the EQS at 2.65 µg/L.

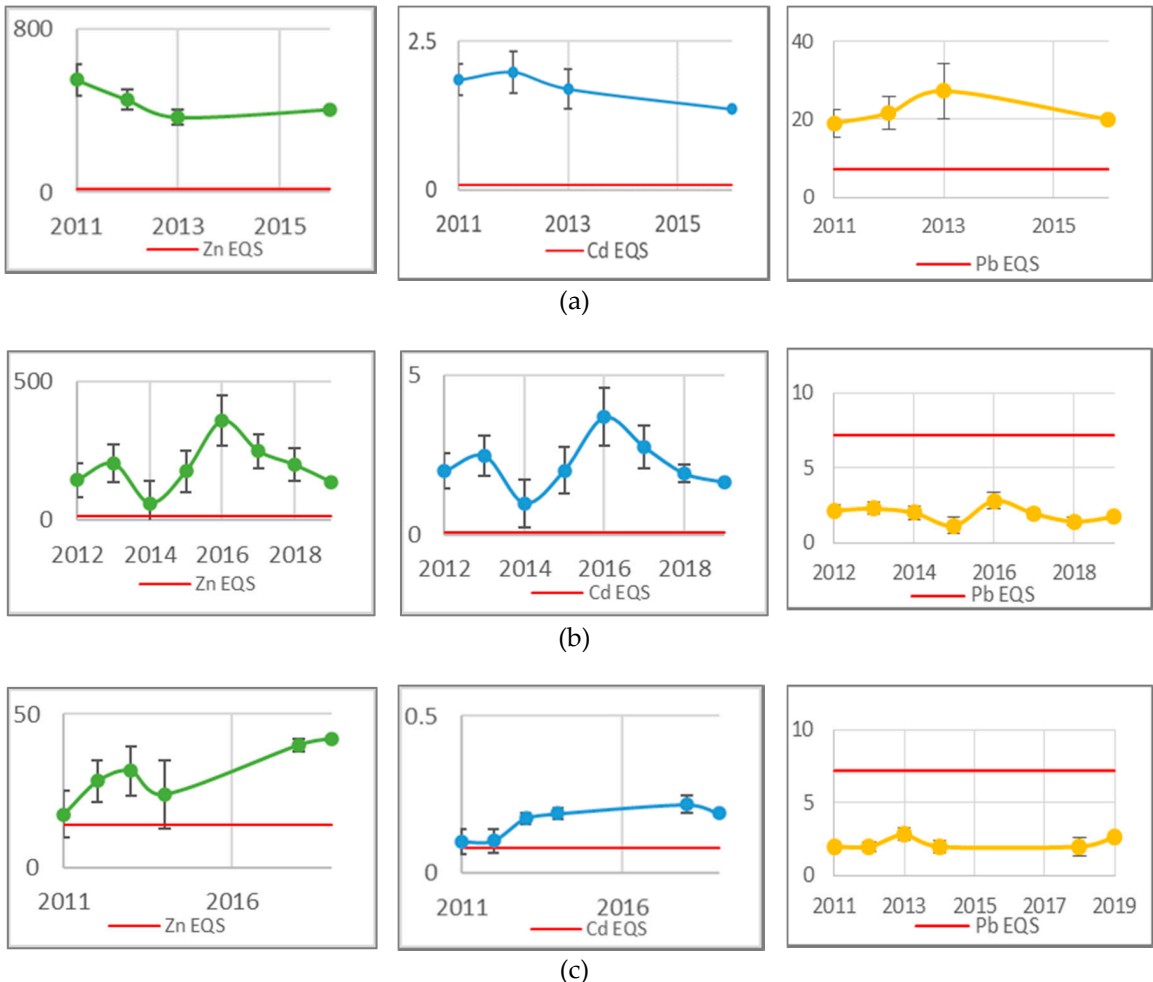

**Figure 7.** Dissolved metal concentrations (μg/L) compared with specific EQS along sample points of the river Teign. (**a**) concentrations of Zn, Cd and Pb at Rookery brook from 2011 to 2016, (**b**) metal concentrations downstream at Beadon brook from 2012 to 2019, (**c**) concentrations from Chudleigh bridge from 2011 to 2019. Data collected from the Environment Agency water quality archive.

Identifying the bioavailable fraction forms the 2nd tier of assessing the risks posed by a pollutant. Concentrations of bioavailable Zn (Cd and Pb) for three sample locations have been calculated with the Biomet tool using DOC, pH and hardness data provided from the EA open data (ESI, Table S4). DOC ranged between 2.8 and 4.7 mg/L, pH ranged from 7.24 to 7.41 and hardness ranged from 24.2 to 37.3 mg/L. Together, these results show that the bioavailable Zn fraction exceeds the EQS at both Chudleigh bridge (18.1 μg/L) and Beadon brook (107 μg/L) (Figure 8). For the river Teign at Preston, bioavailable Zn has stayed closely below the EQS (10.13 μg/L).



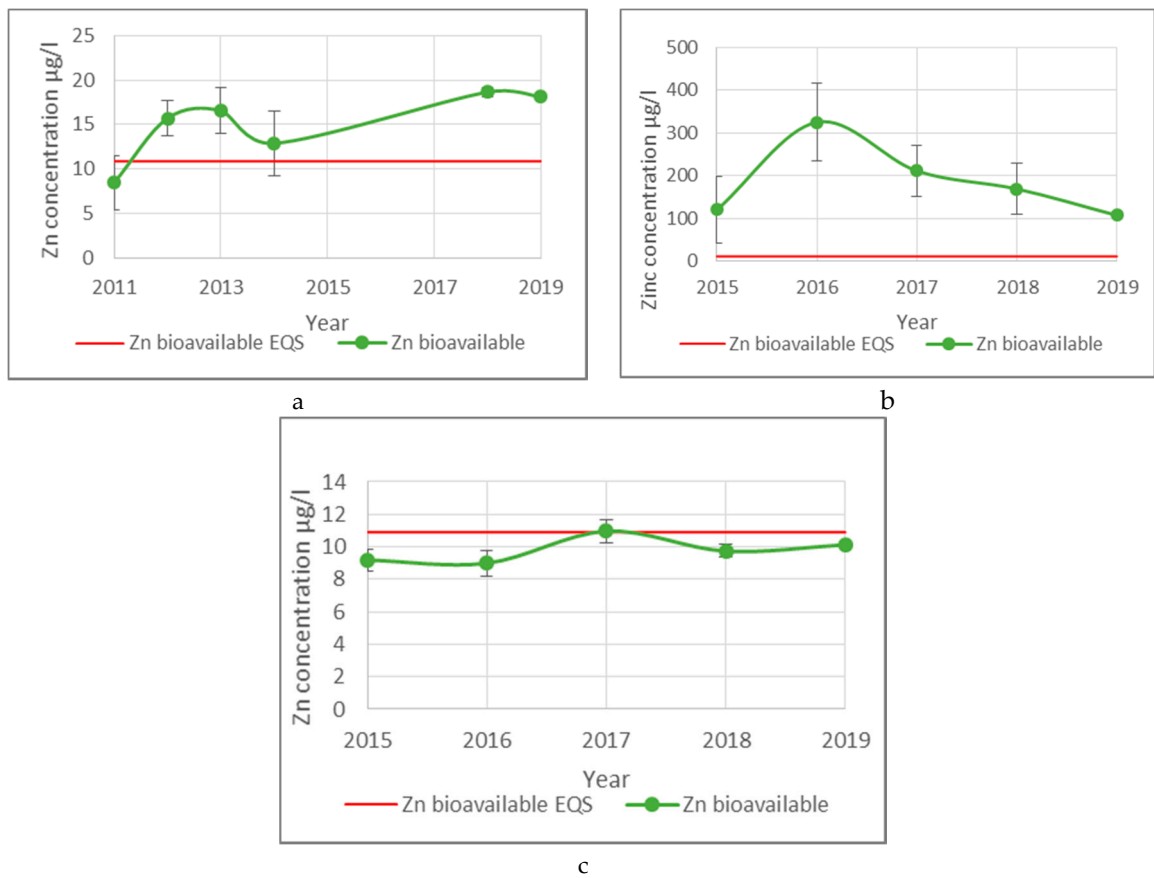

**Figure 8.** Annual mean bioavailable Zn levels for the river Teign at Chudleigh bridge (**a**), beadon Brook (**b**) and Preston (**c**), compared to the bioavailable EQS for Zn. Data calculated from EA water quality archive [8].

### 3.3. Real-World Application Model

The mean bioavailable concentration of Zn at Chudleigh bridge was 18.1 µg/L in 2019. This was combined with flow data at Chudleigh (5.32 m$^3$/s) to estimate the annual load from Bridford downstream to the river Teign; the load was calculated to be 1210 kg/yr. As well as this, the capacity of the pellets was retrieved from adsorption capacity experiments (ESI, Figure S4). The duration of the experiment for the CP and FP was limited, and therefore maximum adsorption capacities were not reached. However, the highest capacities achieved were recorded (Table 3) [19]. For the model, capacities from the field-scale trial were used for a more representative result; the CP were found to have a capacity of 8743 mg/kg for Zn, and 35.40mg/kg for Cd. The most efficient biochar from the experiment was agricultural biochar, with a capacity of 11,000 mg/kg [15], compared to the capacity of the wood biochar, which has been observed at 395.8 mg/kg [16]. The results from the model are shown in Table 4; estimates of the costs of the pellets have been calculated on the basis that 1 tonne of pellets costs £88.95 to dispose of at landfill [21].

The results from the model reveal that the agricultural biochar costs the least amount to reduce Zn levels in the Teign. However, the removal efficiency was only tested up to 2 h, and therefore this is not a realistic value, as the removal efficiency is expected to drop over time. The CP would have a more realistic application, as the field-scale trial showed the removal efficiency to drop to 36% after 3 months in the water; despite this drop, only 383 tonnes of pellets a year would be needed to reduce the Zn levels, costing £34,067. The lower capacity of the FP and lack of removal efficiency data over a longer duration make it an infeasible treatment method costing £717,292 to dispose of at landfill. The wood biochar has the lowest removal efficiency and would require 15,280 tonnes a year, again an infeasible

method. Using the CP, reducing Cd levels below the EQS would require 898 tonnes/yr of pellets, amounting to £79,877; this would cost more than 2-fold the cost of reducing Zn levels. Dissolved Pb concentrations were below the EQS and therefore not considered in the model.

**Table 4.** Table showing the amount of pellets/biochar feedstock required in tonnes/yr to lower Zn levels to the fixed EQS at Chudleigh, based on EA metal concentrations in 2019.

| Treatment Method | Tonnes/yr Required (Assuming 100% Efficiency) | Tonnes/yr Required Based on Removal Efficiencies from This Study | Cost |
|---|---|---|---|
| Compressed Pellet (CP) | 138 | 383 (36% efficiency after 3 months) | £34,067 a year |
| Fired Pellet (FP) | 8064 | 8064 (99.9% efficiency after 53 h) | £717,292 a year |
| Agricultural Biochar | 110 | 137.5 (80% efficiency after 2 h) | £12,230 |
| Wood Biochar | 3056 | 15,280 (20% efficiency after 2 h) | £1,359,156 |

## 4. Discussion

### 4.1. Pellet Removal Efficiency

The data from this study suggest the most suited pellet for removing Zn loads from Bridford are the CP. Fast adsorption rates allow >70% of the metal to be absorbed within the first 2 h of experimentation. This efficiency is supported by the field-scale trial, with rapid adsorption of Zn within the first 20 days (>80%). At the end of the 3 month trial, the removal efficiency drops to <40%, suggesting that the capacity of the pellets had not been exhausted. It was assumed that precipitation of metals as insoluble hydroxides owing to the alkaline pH of the RM pellets or co-precipitation with iron and aluminium oxy-hydroxide floccs becomes the dominant process blocking sorption sites and consequently lowering the adsorption efficiency [26]. The higher removal efficiency of the CP can be attributed to its higher surface area of 27.9 m$^2$/g compared with other pellets (35-fold greater than the FP) [19]. Notably, the FP have a slower initial removal efficiency, yet remove 99.9% of Zn at 53 h. Both pellets cause an immediate pH increase when added to solution, although the FP result in the highest pH increase of 4.68, enabling the formation of precipitates such as iron hydroxide to further drive metal removal. Whilst the adsorption kinetics experiments have identified the CP and FP as having the greatest removal efficiency, the faster removal rate of the CP means a lower phase-contact time is needed between the pellets and water; this makes it more suited to a real-world application. Pb is observed to adsorb more strongly than Zn and Cd, and this is possibly due to its greater partition coefficient [27].

The results from the experiment are supported by other studies using RM pellets [28]. Crushed pellets with a greater surface area (such as the CP) have been found to be most efficient, with enhanced metal adsorption taking place at an optimum pH of 5/6 for Zn. Significant uptake of Zn has been documented within the first few hours of experimentation, with a less pronounced uptake after 24 h [29], in line with the results from this study. Interestingly, the FAE pellets had a lower removal efficiency compared to the CP and PP; however, other studies have proved acid treatment to be highly effective in aiding adsorption [17]. Although surface area is likely to be a key driver in terms of sorption capacity owing to increased sites being available for metal exchange, the charge on the metals of interests as well the adsorbent media themselves will also influence the ability to bind metals. The pH value of the solution in which the pellets are in greatly effects the adsorption and desorption of metal ions. At a low pH, the charge on the outside of the red mud has a high positive charge density, meaning a low uptake of metal ions due to electrostatic repulsion but a high adsorption of anions. However, when the pH increases, the negative charge density on the surface increases, increasing metal adsorption and lowering non-metal adsorption. The presence of Al(OH)$_3$ (gibbsite) and FeO(OH) (goethite), which are hydroxylated surfaces, helps to absorb H$^+$ ions [17] for red media but will have little impact for biochars which exhibit much less variable and more neutral pH.

### 4.2. Biochar Removal Efficiency

Previous studies have demonstrated the remediation potential of Biochar, particularly as a soil modification where application has been seen to reduce bioavailability of toxic metals and simultaneously promote plant growth. Maximum removal efficiencies (>95%) have been observed at high pyrolysis temperatures (650 °C) which greatly influence the success of the treatment method [13]. Other parameters such as contact time, particle size and the type of biochar feedstock used have also been considered as important factors.

Results from this study have shown the type of feedstock to be an important influence on the removal efficiency of Zn, Cd and Pb, rather than pyrolysis temperature. Forestry feedstock had the lowest removal efficiency, whilst agricultural waste had the overall highest. This can be explained by the pyrolysis temperature at which the biochar is produced at. Higher temperatures produce a higher ash content, which raises the pH and consequently aids metal adsorption, with maximum adsorption recorded at pH 5 [14], similarly to the RM pellets. This ash component is accountable for significant Pb immobilisation, explaining why Pb had the greatest adsorption rate in the experiment. Forestry waste has a low ash content and hence low adsorption rates. Other studies support this concept, where lower pH (7.9) has been observed in wood biochars, compared to other feedstock which significantly increases the pH to 9 and above [16].

### 4.3. River Teign Compliance

High metal concentrations do not automatically mean that a water body is failing, and disproportionate results could lead to unnecessary investment in treatment methods to reduce metal concentrations when the toxicity is overestimated. However, dissolved Zn concentrations exceed standards at all sample locations downstream of Bridford mine, suggesting it to be a significant source of Zn to the river Teign. Bioavailable data show that Zn is present in its most ecotoxic form, exceeding the bioavailable EQS all the way downstream to Preston, over 10 km from Bridford mine. Bioavailability data therefore help identify hotspots of high Zn levels such as Beadon Brook and Chudleigh where levels are of environmental concern; the metal is available for biological uptake and present at a concentration that may be harmful to plants and animals. Physiochemical parameters that control the bioavailability of a metal include DOC, pH and hardness. Optimal conditions for bioavailable Zn consist of a DOC ranging between 2.48 and 22.9 mg/L and a pH between 5.7 and 8.4 [30]; results from this study reveal conditions from the Teign at Chudleigh to have a pH of 7.24–7.41, and a DOC ranging between 2.8 and 4.7 mg/L.

Despite this, assessing the compliance of a water body is complex, with many factors to consider. South Devon has naturally high occurring concentrations of heavy metals including Zn owing to its metalliferous background geology. Existing high Zn levels may result in the development of tolerant species that can hyperaccumulate metals [31]. Therefore, the effects of Zn may not be as damaging to ecosystems as studies suggest.

Cd levels in the Teign also exceed the EQS and are rising at some of the sample locations. Independently, the impacts of Cd and resulting effects on ecosystems are beyond the scope of this project. However, the synergistic effects of metals such as Zn, Cd and Pb together have been documented and observed to increase fish mortality [5]. It is therefore important to investigate the effects of combinations of metals to assess the threats posed to the environment.

### 4.4. Application to Bridford Mine

Mine adit drainage tends to be discharged from a single point, as it was the main mechanism of removing water from mines to prevent flooding. In terms of remediation, it is therefore relatively straightforward to divert the flow of mine adit discharges through beds of adsorbent material for passive treatment processes, often using gravity to feed to avoid unnecessary pumping and the requirement of power to the site. The practicalities for application of this treatment at the case study

site (and likely elsewhere) are considered straightforward. According to the real-world application model, the CP and agricultural biochar are the most promising treatment methods for adsorbing Zn at Bridford mine. The removal efficiency of the RM pellets is a result of pre-treatment, which affects the porosity, surface area and adsorption capacity of the pellets. The CP have the highest adsorption capacity due to their larger surface area and would ultimately require less production and lower disposal costs. The FP have a much lower adsorption capacity, resulting in the need for 58-fold more tonnes of pellets a year compared to the CP. Hill (2016) [19] and Turner (2017) [21] similarly found that you would need 44-fold more FP than CP to efficiently remove Zn at the mine site. However, despite the slower adsorption rate of the FP, its ability to significantly raise pH may prove useful for increasing precipitation reactions and consequently removing metals via the formation of hydroxides. A limitation of this study is the difficulty in comparing the mass of pellets required for metal removal when removal efficiency has been measured over different time frames. However, it provides an insight into the potential for the CP and FP to act as an efficient low-cost adsorbent for UK mine sites. The PP also have potential for effectively removing metals at Bridford. However, adsorption capacity data and a field-scale trial would be necessary.

Despite the success of the CP, the field-scale trial by Comber (2015) [20] highlighted a few issues that may affect the pellets' ability to act as an adsorbent. Firstly, the pellets lacked rigidity, resulting in a loss of structural integrity during the trial; this is problematic for a realistic application of the treatment method. Further, the precipitation of ochre (iron hydroxide) resulted in a build up of iron on the pellet surface, blocking adsorption sites, although it also leads to an increase in co-precipitation of other metals, thus limiting the mobility of dissolved metals in the mine water [32].

Field-scale trials of biochar treatment have shown that the effectiveness decreases over time (biochar ageing effect) [33]. However, unlike the CP, biochar is persistent in the environment and its application may be prolonged. This is particularly the case with high-temperature biochars which have a greater carbon stability [34], making it a more effective adsorbent. Biochar therefore offers an attractive remediation alternative to the RM pellets. However, the effects of potentially hazardous substances in biochars, because of the feedstock used and the pyrolysis process, are still largely unknown [35].

Each year, 90 million tonnes of RM are produced globally, making it widely available as an adsorbent [17]. RM as a raw material, although rich in aluminium and iron, does not pose a particular threat to the environment as it binds other metals which might be present as impurities very strongly. Pre-testing of the leaching of metals from the pellets (unpublished data) should have very little desorption into deionised water. Biochars also tend to be relatively inert as organic contaminants are destroyed via the charring process and any residual metal levels are likely to be only very minor impurities. However, once potentially toxic metals have been adsorbed to the medium, it is viewed as a hazardous material and has to be disposed of accordingly, although it may be used within the mine site for land remediation, and off-site disposal is costly, valued at £88.95 per tonne (as at April 2018) [21]. Currently, the pellets can be disposed of in mine tailings in agreement with the EA. However, where this is not possible, they are sent to an inert landfill. One viable solution to reduce disposal costs would be to drain the pellets after use to achieve a greater % of dry weight [19]. Further, to further increase the efficiency of the pellets, a cell-based system could be used, where pellets are placed successively next to each other. This design would ensure that pellet capacity is not all exhausted at once, prolonging their effect of metal removal. Ultimately, recovery of metals from the media and recycling the metal would be the most sustainable option, within a circular economy, but the wholesale value of the trace elements recovered would need to be higher than current market prices for this to be viable. Costs for fabrication of any remediation adsorbent beds, pumping requirements and media purchase were beyond the scope of this study and would also be dependent on the scale of operation, market prices at any given time and pumping requirements.

Adsorption is an economical remediation technique, owing to the abundance of waste materials, their low cost, and high capacities. It is a much more practical option for mine sites than the current most

widely used treatment method activated carbon (AC). AC is inaccessible for most remediation projects due to its high cost, which is typically more than 1000 Euros/tonne, equivalent to £914.281/tonne [36]. The metal-removing capabilities shown by the RM pellets and biochar are therefore more suited to application at Bridford mine than limited methods such as activated carbon.

Realistically, for the river Teign to comply with water quality standards, other inputs need to be addressed. Whilst Bridford mine is a significant source of Zn, it cannot solely be accounted for the failure of Zn levels downstream in the Teign. Other mine inputs such as Wheal Exmouth near Canonteign are a potentially major source of Zn, as shown by the high concentrations at Beadon brook downstream of the mine site. During the peak of mine operation (between 1851 and 1874), outputs of Pb and Zn are estimated to be 11,759 tonnes and 1589 tonnes, respectively [2]. Treatment of mine water at Wheal Exmouth is necessary to reduce metal concentrations below the EQS, particularly in the case of Cd, which would require 898 tonnes of pellets a year applied at Bridford alone to reduce levels below the EQS. Moreover, mine adits only represent point sources of pollution, diffuse sources such as runoff from tailings and road surfaces should also be investigated for their contribution to Zn, Cd and Pb levels.

## 5. Conclusions

This study has highlighted the long-term impact of historical mining on our local water resources, demonstrating the need for protection and assurance of water quality, implemented by key legislation such as the WFD.

The consistent exceedance of Zn and Cd environmental quality standards in the river Teign has formed the rationale for evaluating potential treatment methods. Adsorption techniques for mine remediation are topical due to their low cost and abundance; this study has proven the potential for pelletized RM and biochar as effective adsorbents. Pellets with a greater surface area and higher adsorption capacity such as the CP demonstrate high removal efficiencies for Zn, Cd and Pb within the first 2 h (73.7%, 94.4% and 99.2%, respectively). Agricultural biochar formed at high pyrolysis temperatures has also been observed as a promising material for removing ecotoxic metals (>80% removal within the first 2 h). Limited data on the FAE pellets and PP meant that their application to a mine site could not be determined. However, they do exhibit neutralising capabilities as well as effective adsorption.

Treatment methods need to follow the principle of sustainable development by improving the status of a water body whilst considering the costs and benefits of their application. Reusing the hazardous RM as an adsorptive material supports this concept of sustainability, especially the CP, which can be disposed of at only £34,067; this is much more economically viable than other treatment methods such as activated carbon.

**Supplementary Materials:** The following are available online at http://www.mdpi.com/2075-163X/10/8/721/s1, Figure S1: Steps outlining the kinetics experiment to determine analyte adsorption rates, Figure S2: Steps outlining column experiment, Figure S3: Environment Agency sampling points along the River Teign, Figure S4: Screenshot of Biomet tool, used to calculate the bioavailable concentrations of Zn at Chudleigh bridge using pH, hardness and DOC, Figure S5: Screenshot of Real World Application model, used to calculate the amount of pellets/biochar needed to reduce Zn levels below the EQS at Chudleigh, Table S1: Removal efficiencies for Zn, Cd and Pb at 2, 24 and 53 h, Table S2: Field scale trial at Bridford mine. Percentage removal of Zn, Cd and Pb over a 3-month period, Table S3: Biochar removal efficiency results for Zn, Cd and Pb at 550°C and 700°C, Table S4: Environment Agency monitoring data from 2000–2020.

**Author Contributions:** Conceptualization, S.C.; Methodology S.C., A.T., T.R., A.H.; Investigation A.T., R.H., T.R., A.J.; Formal Analysis A.J., Resources S.C.; Writing A.J.; Supervision S.C. All authors have read and agreed to the published version of the manuscript.

**Funding:** This research received no external funding.

**Acknowledgments:** The authors would like to thank Noel Squibb, owner of Bridford Barytes mine, for his assistance with access and the history of the mine. We would also like to thank Chris Drayson of RMT for providing the test pellets for the red media studies. Finally, we would like to thank Rob Clough and Andy Fisher of the University of Plymouth for their support with the analytical aspects of this research.

**Conflicts of Interest:** The authors declare NO conflict of interest associated with this work financially, personally or otherwise.

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
