# Peer review of "Assessing Options for Remediation of Contaminated Mine Site Drainage Entering the River Teign, Southwest England"

_minerals, doi:10.3390/min10080721_

Round 1

Reviewer 1 Report

The manuscript requires major revision to merit publication.

(1) the novelty of the work needs to be highlighted in the abstract and Introduction.

(2) the research gaps need to be clarified in the Introduction. How your work differs from previous research e.g. Bertocchi et al. 2005 01 Dec 2005, 134(1-3):112-119. Biochar has not been mentioned in the introduction.

(3) Methodology requires clarification. It seems the paper brings together results of 'unpublished' reports and dissertations. Some parts of the methodology contains background information of these work rather than the methods. Those dissertations may not be available online so add detail of the used methods and materials. When samples were collected? How many sampling events? Adsorption kinetics and titration capacity experiments need to be described. What is the solid-liquid used for adsorption tests?  Were analyses done in triplicates? How metal concentration was measured, cite equipment maker and model. What are the characteristics of the adsorbents e.g. grain size, porosity,  and surface area? Do you have SEM analyses? Provide more information about the Real-World Application Model. What was the volume of river water used for simulation?

(4) Results section should not contain discussion e.g. lines 386-388 should be in the Discussion. Section 3.2 should come first as 3.1. Some figures need to be improved and figure caption clarified by adding the initial concentration of the metals. Figure 7 is a repetition of Figure 1, please delete it. Figure 3 seems to be missing parts e.g. A and C missing 25 hour information, and B and D missing right frame; and abbreviations in the figure caption need to be defined. For consistency use the abbreviations as in Table 1. Figure 5 - no need to repeat title inside figure and check axe's format.

(5) There is room for improving discussion. You need to explain better your results, providing justification for e.g. line 444 - one adsorbent being better than the other. Could be due to surface area or surface charges?  Also, you need to explore how you will apply these adsorbent in practice? Will you divert the river to a reactor? You mentioned RM is toxic, could this adsorbent leak toxic compounds to the river water? In terms of costs, what are the costs of acquiring the RM and Biochar? Disposing the used adsorbents in the landfill is not a sustainable approach since they may leak heavy metals. Did you check the desorption rate? Or did you explore any possibility of recovering the metals from these adsorbents? Discussion around these points will improve the work.

(6) Conclusions must contain some numerical results.

Author Response

Thank you for your considered review. We have made the recommended changes and detailed them in the attached document (summary of all reviews for clarity). 

Reviewer 2 Report

Review of Minerals 879575 : Assessing option for remediation of contaminated mine site drainage entering the River Teign, Southwest England

The article presents a summary of several studies done on treatment of Zn and Cd contaminated water using adsorption of red mud pellets and biochar. The topic is of interest to the scientific community and the research offers some interesting insights. However, I feel the article tried to summarize too much the research, and skipped important information that should be included in the main article to present a complete research. For this reason, I suggest major revision before accepting the paper. Specific comments are exposed below.

Line 75: Is the classification based on single or periodic sampling campaigns?

Line 99: Please provide more specific examples of red media usage for adsorption. Type of contaminants, concentration range, performance achieved with and without pretreatment, etc. Some of this information is presented in sections 2.2 and 2.3, which should be transferred to the introduction.

Figure 1: Units of Zn concentration?

Sections 2.2 and 2.3: Place in introduction, or add a subsection in introduction (e.g. background studies); this information does not belong in methodology.

Line 192: More info on methodology can be added in a few sentences, instead of referring to supplementary material. What volume of water sample was used with 200g of pellets? Was it agitated? For how long? How many samples taken, at which frequency? Or place figure S1 in the main text.

Line 205: Here too, add info on the experimental procedure.

Section 2.5: The biochar should be introduced in introduction, and previous studies on the adsorption capacity relevant to the case study should be presented.

Section 2.7: Add some info, explain the calculation. The image in S10 is too compressed to be of use, the equations are more relevant than the screen shot.

S2: Where are Table 14 and Figure 18? In the third box, verify the average volume, it is reported as a mass.

Figure 3: Points at 24 hours are not visible in A, B, C.

Line 260: No need for S4, it is the same data as Figure 3.

Line 274: Why were the FAE and PP tests stopped after 6 hours? In Figure 3 there are data points at 24 hours.

Figure 4: The field test needs to be explained more. Was the water flowing through the pellets at a fixed flow rate? Was the initial concentration stable, or the mine water quality varied in the 3-month period? How does the water to solid ratio compare to the laboratory tests? Before field trial, please present the results of the adsorption capacity column tests.

Figure 5: Do these results come from laboratory or field tests? What is the relevance of 2 hours, when the kinetic test showed better results after 24 hours?

Figure 7: Remove, same info is found in Figure 1

Table 3: How can you compare the mass required when efficiencies are measured over very different time frames? For example, can FP keep a removal efficiency of 99.9% over a year? And why not include also cost of fabrication (for pellets and biochar)? Please add a column indicating predicted Zn concentration after treatment for each treatment method.

Line 430: Add information on precipitation: which species precipitated? How was it observed (visual, microscopy)?

Reference 7: Anne-Marie and Emmanuelle are first names, correct the reference.

Author Response

(The authors gave the same response as above.)

Round 2

Reviewer 2 Report

Most of the comments were answered in a satisfactory way. The methodology is much improved, the results section flows better. The discussion was also improved.